# Temporal variation in food consumption of Brazilian adolescents (2009-2015)

**Hélida Ventura Barbosa Gonçalves**[1ම], **Daniela Silva Canella**[2ම], **Daniel Henrique Bandoni**[1ම]*

1 Instituto de Saúde e Sociedade, Universidade Federal de São Paulo, Santos, SP, Brazil, 2 Instituto de Nutrição, Universidade do Estado do Rio de Janeiro, Rio de Janeiro, RJ, Brazil

ම These authors contributed equally to this work.
* danielbandoni@gmail.com

## Abstract

### Background

Worldwide population has been increasingly exposed to ultra-processed foods, which are associated with obesity. Adolescence is a transition period of life and WHO recommends the surveillance of risk factors to the adolescents' health, such as diet, because experiences in this phase can result in health risks.

### Objective

To assess the trends in food consumption of adolescents from Brazilian capitals according to sociodemographic variables, based on data from the National Survey of School Health (PeNSE).

### Methods

Data from in 2009, 2012 and 2015 of a total of 173,310 9th graders enrolled in public and private schools in Brazilian capitals and in the Federal District were assessed. Food consumption was assessed from regular consumption (five or more times a week) of healthy eating markers (beans; vegetables; fruit) and unhealthy eating markers (sweets; soft drinks; fried salty snacks). For sociodemographic variables, we considered macro regions; age; race/ skin color; gender; school administrative status. We assessed these markers trends for the population and, additionally, the analyses were stratified by gender, race/ skin color, and school administrative status. Statistical significance of the temporal trends was assessed by linear regression model.

### Results

Over six years, three types of change in Brazilian adolescents' diet were observed: decreasing regular consumption of beans, sweets and soft drinks, increasing regular consumption of vegetables, and stable consumption of fruit and fried salty snacks.

**Data Availability Statement:** For data analysis of this study, we used the databases of the 2009, 2012 and 2015 editions of the National School Health Survey. Available at the following links:

2009: https://www.ibge.gov.br/estatisticas/downloads-estatisticas.html?caminho=pense/2009/microdados/ 2012: https://www.ibge.gov.br/estatisticas/downloads-estatisticas.html?caminho=pense/2012/microdados/ 2015: https://www.ibge.gov.br/estatisticas/downloads-estatisticas.html?caminho=pense/2015/microdados/.

**Funding:** The authors received no specific funding for this work.

**Competing interests:** The authors have declared that no competing interests exist.

## Conclusion

Brazilian adolescents' diet composition has changed in a short period, and therefore it is necessary to monitor it to propose actions aimed at this public.

## Introduction

Adolescence is the transition period between childhood and adulthood, and, according to the World Health Organization (WHO), 10 to 19 are the age limits that define adolescence. This population represents about 18% of the world's population and approximately 90% of them live in low-income countries. In Brazil, adolescents account for almost 20% of the total population [1–5].

Worldwide, 20% to 25% of adolescents are overweight, and the number of children and adolescents with obesity has increased 10 times in the last 40 years [6]. In Brazil, 20.5% of this population is overweight and 4.9% has obesity [7].

The increasing prevalence of overweight is associated with the change in the population's food profile, with an increasing consumption of ultra-processed foods and a decreasing consumption of fresh or minimally processed foods [8, 9].

Ultra-processed foods are those resulting from a series of industrial processes. Generally, colors, flavors (sugar, oils, fats and salt), emulsifiers and other additives are added to them to improve the taste. These points together with sophisticated packaging make the product attractive to consumers, especially children and adolescents [10].

Adolescents are considered the group of higher risk in relation to diet, because they have higher frequency of fast food consumption, sugary drinks, biscuits and salty food, as well as sedentary behaviors, such as staying long periods in front of the television, computers and video games, which favors this unhealthy eating habit [9, 11–13].

Considering this scenario, monitoring adolescents' health is important [14] and WHO recommends implementing and maintaining systems for surveillance of risk factors to the adolescents' health, because experiences in this phase can result in health risks in the present and in the future [15–17]. Monitoring the health of adolescents is already studied in international research such as the Global School Based Student Health Survey (GSHS), but little was known about this condition in Brazil. To this end, in Brazil, since 2009 the National Survey of School Health Survey (*Pesquisa Nacional de Saúde do Escolar*—PeNSE) has been carried out, which in addition to monitoring the health of adolescents, also aims to provide information for the planning of public policies, as well as to subsidize managers with information, and thus sustain the surveillance system for schoolchildren [18, 19].

School is an important environment for health promotion, and food supply to schoolchildren is one of the strategies for this to be effective. In Brazil, the National School Feeding Program (*Programa Nacional de Alimentação Escolar*—PNAE), a public policy that offers free of charge meals to students from public schools, aims to partially meet nutritional requirement during school period and, consequently, improve learning capacity [20]. School feeding positively affects the consumption of healthy foods [21].

Therefore, this study aims to assess the trends of food consumption by adolescents from Brazilian capitals, according to sociodemographic variables, based on data from the National Survey of School Health from 2009, 2012 and 2015.

## Materials and methods

### Database and sample

This is a temporal trend study using data from three cross-sectional surveys, the National Survey of School Health (PeNSE), conducted in 2009, 2012 and 2015 [3, 19, 22]. The research

project was developed by an agreement between the Brazilian Institute of Geography and Statistics (IBGE) and the Ministry of Health, with the support of the Ministry of Education, which aimed to investigate behavioral risk and health protection factors in adolescent students. The research data are public and are available through the IBGE website (https://www.ibge.gov.br/estatisticas/sociais/saude/9134-pesquisa-nacional-de-saude-do-escolar.html?=&t=microdados).

PeNSE is a school-based epidemiological research that had as study population a group of schoolchildren who attended the 9th grade of elementary school, of public and private schools in Brazil. The three samples (2009, 2012, 2015) that comprised the present study were independent samples, as shown in Fig 1.

PeNSE 2009 sample was representative of the all Brazilian capitals and the Federal District (n = 27). In the 2012 and 2015 editions, in addition to representing the capitals and the Federal District, it was possible to represent them in large geographic regions and Federation Units, respectively.

PeNSE uses a probabilistic sample planned as follows: definition of geographic strata, draw of schools and, subsequently, of 9th grade classes; in the classes drawn, assessment of the schoolchildren that accepted to participate in the survey. Geographical stratification occurred as follows, in each of the studies: 1) 2009: each of the 26 municipalities of the capitals and the Federal District was considered as a geographical stratum, totaling 27 geographic strata; 2) 2012: the 2009 stratification was also performed in 2012, but students from cities belonging to non-capitals were also assessed, and these municipalities in each region were grouped into a stratum, totaling 32 geographical strata; 3) 2015: the stratification was expanded with non-capital municipalities, which were grouped into a stratum for each of the Federation Units. Therefore, this edition had 53 geographical strata.

For this study, only representative data of the capitals and the Federal District (DF) were used for comparison between the three years of study (n = 173,310). Thus, in each of the 27 capitals

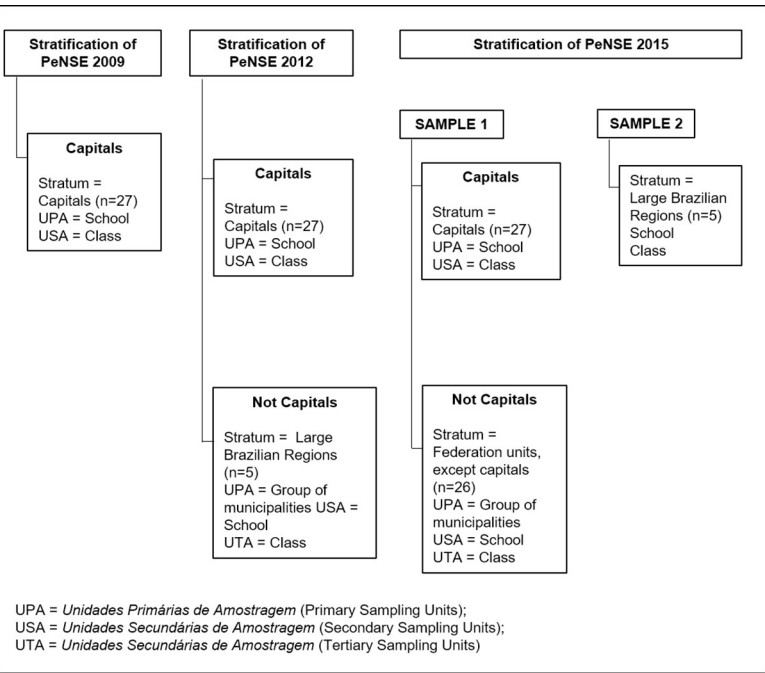

**Fig 1. Stratification of PeNSE.**

strata, a sample of schools was selected, and, in each school, we selected a sample of classes. Samples belonging to non-capitals and data from PeNSE 2015 sample 2 were excluded from this analysis. PeNSE, in the three editions, excluded, from the answer to the questionnaires, schools without 9th grade, schools with less than 15 students in the 9th grade, disabled participants, school not visited and refusal to participate. All students from the selected classes present on the day were invited to answer the questionnaire. In the three samples, those who agreed to participate in the research and with the informed consent form participated in the study [3, 18, 21].

A total of 1,453 schools and 2,175 classes comprised the 2009 sample. Regarding the number of students, 63,411 were present on the collection day, and 60,973 (96.15%) were considered valid questionnaires—those of students who reported they would like to participate in the study and who had filled the information about gender and age. A total of 1,469 schools, 2,219 classes and 74,436 (98.4% of valid questionnaires) enrolled students comprised the 2012 capital sample. In 2015, 51,303 valid questionnaires were answered, from 1,966 classes belonging to 1,339 schools.

The projects developed for each edition of PeNSE were submitted and approved by the National Research Ethics Commission (Comissão Nacional de Ética em Pesquisa—CONEP).

## Data collection and study variables

Collection of data from the three editions of PeNSE occurred throughout 2009, 2012 and 2015. PeNSE was composed of two collection instruments: a questionnaire applied with the unit principal, related to general school questions, and a self-administered questionnaire, structured in thematic modules, which students answered, in 2009 via Personal Digital Assistant (PDA) and via smartphone in 2012 and 2015. In this study, the modules food consumption and sociodemographic characteristics were used. Electronic devices were made available by IBGE researchers on the day of interviews.

The main variables of interest in this study are the regular consumption of healthy and unhealthy eating markers foods in the three surveys. Information on food consumption refers to frequency of consumption considering the whole day's food, during the seven-day period, including consumption at home, at school, and in other environments. The data was obtained, as in this example: in the last 7 days, on how many days did you eat fresh fruit or fruit salad?, in which the possible answers were: I did not eat; 1 day; 2 days; 3 days; 4 days; 5 days; 6 days; every day; uninformed. The results were recategorized and individuals without information were excluded from our analyses.

Food items collected were grouped in:

- Healthy eating markers: fruit; vegetables; beans.

- Unhealthy eating markers: fried salty snacks (such as chicken croquette, fried beef croquette, fried *pastel*, *acarajé* etc.); sweets (such as sweets, candies, chocolate, chewing gums, bonbons, or lollipops); and soft drinks.

Consuming these foods five or more times a week was considered regular.

For sociodemographic variables, we considered: Brazilian macro regions (North, Northeast, Southeast, South, Midwest); age (<13, 13, 14, 15 and >16 years of age); race/skin color (white, black, Asian, mixed race and indigenous); gender (female and male); and schools administrative status (public and private).

## Data analysis

The distribution of students according to sociodemographic characteristics in the three years of survey was described by frequency and confidence interval (95%CI).

The frequency of students with regular consumption (five or more times a week) of each of the six eating markers in 2009, 2012 and 2015, as well as the confidence intervals (95%CI), was described for the population studied each year and according to sociodemographic characteristic.

The statistical significance of the indicator's temporal trends was assessed using a linear regression model, with the indicator value as an outcome (dependent variable)—e.g., the percentage of adolescents who regularly consume soft drinks—and the year of the survey (2009, 2012 or 2015) as an explanatory variable, both expressed as continuous variables. The regression coefficient of the model indicates the average annual rate, expressed in percentage points per year, of indicator variation in the period. All models were adjusted by sociodemographic characteristic (gender, age, macro regions, race/skin color, and school administrative status). The variation corresponding to a regression coefficient statistically different from zero (p value $\leq$ 0.05) was considered significant.

The complex design of the sample is considered through the weighting that was applied, in order to obtain representative estimates of the intended population. For this purpose, the survey command (svy) was used in the software Stata version 13.1 (StataCorp LP, College Station, USA).

## Results

Table 1 shows the distribution of the population's sociodemographic characteristic in the three surveys. The population aged 14 years old increased (from 39.14% in 2009 to 41.22% in 2015). In contrast, fewer adolescents defined themselves as white (from 40.15% to 36.51%).

Table 2 shows the variation in regular consumption of healthy and unhealthy eating markers among students. Vegetable consumption (mean variation 5.100) was the only healthy eating marker that positively evolved, while the frequency of regular bean consumption reduced (−2.933). Among foods that are markers of an unhealthy diet, the fall in consumption of sweets and soft drinks must be highlighted, which was nevertheless highly frequent.

Table 3 shows trends was similar in both genders. Bean consumption declined more among girls, who already had a lower prevalence of regular consumption in 2009 (−3.565). Despite having similar trends, regular sweet consumption was higher in girls, still close to 50% in 2015.

Table 4 shows the eating markers according to race/skin color. The changes in consumption, although similar, contributed to approximate regular consumption between the categories in the different markers in the 2015 survey. Sweet consumption declined more strongly among white adolescents (−8.580), while soft drink consumption declined more among mixed–race students (−6.572).

Table 5 shows the results for students from public and private schools. In private schools, the consumption of sweets (−9.291) and soft drinks (−7.112) fell. Bean consumption was higher in public schools in the three years, but with a higher negative variation in the period, whereas fruit consumption negatively varied only for adolescents in private schools.

## Discussion

The variation in food consumption by Brazilian adolescents shows a positive evolution, increasing vegetable consumption and decreasing sweet and soft drink consumption, but the habit of regularly consuming beans also decreased. Food consumption by the Brazilian population in general has undergone changes in recent decades, decreasing the consumption of basic foods, such as rice and beans, increasing the consumption of ultra-processed foods, maintaining the consumption of sugary products and insufficiently consuming fruit and vegetables [23].

**Table 1. Sociodemographic characteristics in Brazilian adolescents from 2009 to 2015.** Brazil, 2009, 2012 and 2015.

| Variable | 2009 | | | 2012 | | | 2015 | | |
|---|---|---|---|---|---|---|---|---|---|
| | % | 95% CI | | | 95% CI | | % | 95% CI | |
| | | Min | Max | | Min | Max | | Min | Max |
| **Macro regions** | | | | | | | | | |
| North | 11.23 | 11.13 | 11.33 | 11.69 | 11.01 | 12.40 | 12.81 | 11.98 | 13.69 |
| Northeast | 23.9 | 23.76 | 24.04 | 23.67 | 22.70 | 24.66 | 23.83 | 22.61 | 25.11 |
| Southeast | 46.96 | 46.74 | 47.19 | 45.15 | 43.59 | 46.71 | 44.64 | 42.89 | 46.41 |
| South | 6.81 | 6.75 | 6.86 | 7.15 | 6.62 | 7.73 | 6.03 | 5.46 | 6.65 |
| Mid-west | 11.10 | 11.02 | 11.18 | 12.34 | 11.64 | 13.08 | 12.69 | 11.85 | 13.57 |
| Total (n) | 60,973 | | | 61,145 | | | 51,192 | | |
| **Gender** | | | | | | | | | |
| Male | 47.46 | 46.77 | 48.16 | 49.16 | 48.42 | 49.90 | 49.21 | 48.33 | 50.09 |
| Female | 52.54 | 51.84 | 53.23 | 50.83 | 50.10 | 51.58 | 50.79 | 49.90 | 51.67 |
| Total (n) | 60,973 | | | 61,145 | | | 51,192 | | |
| **Race/skin color** | | | | | | | | | |
| White | 40.15 | 39.47 | 40.82 | 37.72 | 36,16 | 39.31 | 36.51 | 34.75 | 38.32 |
| Black | 12.87 | 12.42 | 13.33 | 14.19 | 13,46 | 14.95 | 13.51 | 12.73 | 14.32 |
| Asian | 3.75 | 3.50 | 4.01 | 4.50 | 4.21 | 4,81 | 5.04 | 4.71 | 5.40 |
| Mixed race | 39.14 | 38.48 | 39.80 | 39.91 | 38.61 | 41,22 | 41.75 | 40.27 | 13.24 |
| Indigenous | 4.09 | 3.85 | 4.35 | 3.68 | 3.44 | 3,94 | 3.19 | 2.96 | 3.45 |
| Total (n) | 59,805 | | | 61,114 | | | 51,144 | | |
| **Age** | | | | | | | | | |
| <13 | 0.71 | 0.63 | 0.80 | 0.49 | 0.41 | 0.61 | 0.34 | 0.26 | 0.45 |
| 13 | 23.74 | 23.15 | 24.34 | 18.60 | 17.72 | 19.51 | 20.00 | 18.95 | 21.11 |
| 14 | 47.09 | 46.39 | 47.78 | 50.26 | 49.18 | 51.33 | 52.43 | 51.29 | 53.57 |
| 15 | 18.25 | 17.73 | 18.77 | 19.41 | 18.54 | 20.32 | 18.48 | 17.52 | 19.47 |
| >16 | 10.22 | 9.84 | 10.61 | 11.23 | 10.40 | 12.12 | 8.74 | 8.09 | 9.45 |
| Total (n) | 60,806 | | | 60,930 | | | 51,192 | | |
| **School administrative status** | | | | | | | | | |
| Private | 20.80 | 20.70 | 21.00 | 25.49 | 21.86 | 29.50 | 27.11 | 23.19 | 31.44 |
| Public | 79.20 | 79.04 | 79.35 | 74.51 | 70.50 | 78.14 | 72.89 | 68.57 | 76.81 |
| Total (n) | 60,973 | | | 61,145 | | | 51,192 | | |

According to data from the Surveillance System of Risk and Protective Factors for Chronic Diseases by Telephone Survey (Sistema de Vigilância de fatores de risco e proteção para

**Table 2. Trends in prevalence of regular food consumption (≥ 5 times/ week) of eating markers from 2009 to 2015.** Brazil, 2009, 2012 and 2015.

| Eating markers | 2009 | | | 2012 | | | 2015 | | | Average variation | p value |
|---|---|---|---|---|---|---|---|---|---|---|---|
| | % | 95% CI | | % | 95% CI | | % | 95% CI | | | |
| **Beans** | 62.55 | 61.91 | 63.18 | 60.00 | 58.47 | 61.50 | 56.26 | 54.89 | 57.62 | -2.933 | <0.001 |
| **Vegetables** | 31.23 | 30.58 | 31.89 | 35.88 | 35.07 | 36.70 | 38.27 | 37.30 | 39.25 | 5.100 | <0.001 |
| **Fruits** | 31.51 | 30.85 | 32.18 | 29.76 | 29.08 | 30.46 | 32.82 | 31.95 | 33.70 | 0.149 | 0.557 |
| **Fried salty snacks** | 12.48 | 12.04 | 12.93 | 15.74 | 15.15 | 16.35 | 14.46 | 13.74 | 15.21 | 0.757 | <0.001 |
| **Sweets** | 50.87 | 50.17 | 51.57 | 42.62 | 41.65 | 43.59 | 41.78 | 40.91 | 42.64 | -7.410 | <0.001 |
| **Soft drinks** | 37.21 | 36.53 | 37.90 | 35.44 | 34.64 | 36.25 | 28.84 | 27.86 | 29.83 | -4.407 | <0.001 |

*adjusted for the variables age, gender, race/skin color, macro regions and school administrative status.

**Table 3. Trends in prevalence of regular food consumption (≥ 5 times/ week) of eating markers, according to gender, from 2009 to 2015.** Brazil, 2009, 2012 and 2015.

| Eating markers | 2009 | | | 2012 | | | 2015 | | | Average variation | |
|---|---|---|---|---|---|---|---|---|---|---|---|
| | % | 95% CI | | % | 95% CI | | % | 95% CI | | | *p* value |
| **Male** | | | | | | | | | | | |
| **Beans** | 68.27 | 67.38 | 69.14 | 65.92 | 64.24 | 67.56 | 61.18 | 59.68 | 62.65 | -2.888 | <0.001 |
| **Vegetables** | 31.20 | 30.26 | 32.15 | 35.58 | 34.59 | 36.58 | 37.93 | 36.59 | 39.29 | 5.933 | <0.001 |
| **Fruits** | 31.43 | 30.45 | 32.42 | 29.85 | 28.83 | 30.89 | 33.33 | 32.24 | 34.44 | 0.187 | 0.610 |
| **Fried salty snacks** | 12.34 | 11.70 | 13.00 | 14.60 | 13.77 | 15.48 | 13.70 | 12.88 | 14.56 | 0.237 | 0.389 |
| **Sweets** | 42.58 | 41.56 | 43.62 | 36.22 | 34.86 | 37.61 | 35.99 | 34.78 | 37.22 | -7.002 | <0.001 |
| **Soft drinks** | 37.89 | 36.88 | 38.90 | 36.46 | 35.37 | 37.56 | 30.48 | 29.27 | 31.73 | -4.842 | <0.001 |
| **Female** | | | | | | | | | | | |
| **Beans** | 57.41 | 56.51 | 58.29 | 54.28 | 52.60 | 55.95 | 51.50 | 49.80 | 53.19 | -3.565 | <0.001 |
| **Vegetables** | 31.26 | 30.38 | 69.62 | 36.18 | 35.21 | 37.15 | 38.61 | 37.47 | 39.75 | 6.357 | <0.001 |
| **Fruits** | 31.59 | 30.69 | 32.50 | 29.68 | 28.75 | 30.62 | 32.33 | 31.17 | 33.51 | 0.189 | 0.649 |
| **Fried salty snacks** | 12.60 | 12.01 | 13.22 | 16.84 | 16.09 | 17.62 | 15.19 | 14.22 | 16.22 | 1.815 | <0.001 |
| **Sweets** | 58.33 | 57.41 | 59.25 | 48.80 | 47.62 | 49.97 | 47.38 | 46.31 | 48.44 | -7.496 | <0.001 |
| **Soft drinks** | 36.61 | 35.69 | 37.54 | 34.46 | 33.37 | 35.56 | 27.24 | 26.11 | 28.40 | -3.724 | <0.001 |

*adjusted for the variables age, race/skin color, macro regions and school administrative status.

doenças crônicas por inquérito telefônico—VIGITEL), regular bean consumption reaches 66% Brazilian adults. Men had a higher consumption percentage (73%), while among women consumption reached 61% [24]. We can verify that in our study the adolescents' consumption was lower than that of the Brazilian adult population, maybe due to the feeding practices of this phase of life, influenced by family practices, social cycle, socioeconomic conditions, and the influence of the media [15].

Sweet and soft drink consumption has also decreased over the three surveys. However, it is noteworthy that although they decreased, these percentages are still high. A study conducted in the city of São Paulo revealed that the higher the consumption of sugary drinks the worse the quality of the diet, with lower consumption of fruit, vegetables, meat and eggs [25]. Sugary drinks constitute over 13% of the average daily energy consumption among U.S. children and adolescents [26]. The consumption of sugary drinks and sweets may be due to the excess of advertisements, which reach mainly children and adolescents [27].

The school administrative status can be considered a proxy for family income, and in this study, a higher sweet consumption was verified among adolescents who studied in public schools. However, the consumption of ultra-processed foods presents a growing participation in the Brazilians' diet, in all income ranges. The fact that schools have cafeterias and/or nearby establishments also influences this consumption. It is worth noting that in Brazil there is no specific national legislation that prohibits the sale of processed foods in or near schools. There are only state or municipal initiatives. Therefore, assessing the school food environment is important, as it will interfere in the quality of meals, as well as in the students' adherence to the school food [23, 28, 29, 30].

In a study that analyzed characteristics of the food environment in Brazilian schools, a high frequency of sales of ultra-processed foods in schools was identified. This research also noted that schoolchildren who receive meals in schools have 35% less chance of obesity when compared to the others. The fact is that not all Brazilian schoolchildren receive meals at schools. Only students enrolled in public schools have their food subsidized by the federal government, planned by nutritionists. Even so, adherence is low. Students enrolled in private schools pay

**Table 4. Trends in prevalence of regular food consumption ($\geq$ 5 times/ week) of diet markers, according to race/skin color, from 2009 to 2015.** Brazil, 2009, 2012 and 2015.

| Eating markers | 2009 | | | 2012 | | | 2015 | | | Average variation | p value |
|---|---|---|---|---|---|---|---|---|---|---|---|
| | % | 95% CI | | % | 95% CI | | % | 95% CI | | | |
| **White** | | | | | | | | | | | |
| Beans | 61.02 | 59.93 | 62.1 | 57.61 | 55.45 | 59.74 | 54.14 | 52.14 | 53.12 | -3.553 | <0.001 |
| Vegetables | 33.41 | 32.31 | 34.54 | 38.49 | 37.16 | 39.84 | 40.93 | 39.32 | 42.57 | 6.172 | <0.001 |
| Fruits | 32.97 | 31.84 | 31.11 | 30.30 | 28.92 | 31.71 | 33.48 | 31.87 | 35.13 | 0.497 | 0.242 |
| Fried salty snacks | 11.23 | 10.56 | 11.93 | 14.48 | 13.60 | 15.41 | 13.29 | 12.15 | 14.51 | 0.503 | 0.115 |
| Sweets | 49.75 | 48.57 | 50.92 | 40.42 | 39.04 | 41.81 | 40.60 | 39.37 | 41.84 | -8.580 | <0.001 |
| Soft drinks | 37.94 | 36.79 | 39.10 | 36.01 | 34.77 | 37.27 | 28.42 | 27.00 | 29.87 | -5.531 | <0.001 |
| **Black** | | | | | | | | | | | |
| Beans | 66.5 | 64.85 | 68.11 | 64.29 | 62.23 | 66.31 | 59.84 | 57.13 | 62.48 | -3.526 | <0.001 |
| Vegetables | 28.3 | 26.59 | 30.08 | 34.51 | 32.69 | 36.37 | 36.17 | 34.35 | 38.02 | 6.544 | <0.001 |
| Fruits | 30.45 | 28.65 | 32.31 | 29.92 | 28.18 | 31.72 | 32.77 | 31.07 | 34.51 | 0.874 | 0.194 |
| Fried salty snacks | 14.15 | 12.82 | 15.59 | 17.78 | 16.31 | 19.36 | 17.20 | 15.53 | 19.01 | 1.361 | 0.011 |
| Sweets | 50.04 | 48.11 | 51.98 | 43.52 | 41.55 | 45.51 | 43.79 | 41.62 | 45.97 | -4.625 | <0.001 |
| Soft drinks | 38.49 | 36.61 | 40.41 | 36.91 | 35.24 | 38.61 | 32.30 | 30.18 | 34.52 | -3.355 | <0.001 |
| **Asian** | | | | | | | | | | | |
| Beans | 63.24 | 62.27 | 64.21 | 57.40 | 54.29 | 60.46 | 53.31 | 49.94 | 56.65 | -5.390 | 0.888 |
| Vegetables | 29.92 | 28.93 | 30.93 | 36.56 | 33.33 | 39.92 | 41.04 | 37.08 | 45.11 | 5.912 | <0.001 |
| Fruits | 30.82 | 29.79 | 31.86 | 30.12 | 27.36 | 33.04 | 33.06 | 29.60 | 36.71 | 0.642 | 0.390 |
| Fried salty snacks | 12.90 | 12.21 | 13.63 | 17.34 | 15.44 | 19.42 | 18.88 | 15.57 | 22.70 | 2.892 | <0.001 |
| Sweets | 52.75 | 51.67 | 53.83 | 46.51 | 43.47 | 49.57 | 44.37 | 41.05 | 47.75 | -2.499 | 0.002 |
| Soft drinks | 36.31 | 35.26 | 37.37 | 32.94 | 30.06 | 35.95 | 28.62 | 25.32 | 32.15 | -3.924 | <0.001 |
| **Mixed race** | | | | | | | | | | | |
| Beans | 57.96 | 54.71 | 61.14 | 61.11 | 59.60 | 62.59 | 57.21 | 5576 | 58.65 | 0.564 | 0.542 |
| Vegetables | 32.33 | 29.16 | 35.68 | 34.09 | 33.16 | 35.02 | 36.41 | 35.26 | 37.56 | 6.637 | <0.001 |
| Fruits | 30.25 | 27.13 | 33.57 | 29.16 | 28.17 | 30.16 | 32.40 | 31.25 | 33.57 | 0.955 | 0.267 |
| Fried salty snacks | 13.43 | 11.34 | 15.84 | 15.83 | 15.03 | 16.66 | 14.08 | 13.34 | 14.86 | 0.933 | 0.171 |
| Sweets | 51.31 | 47.86 | 54.75 | 44.24 | 42.94 | 45.55 | 42.07 | 40.98 | 43.16 | -3.514 | <0.001 |
| Soft drinks | 35.04 | 31.81 | 38.41 | 34.69 | 33.59 | 35.81 | 28.04 | 26.89 | 29.23 | -6.572 | <0.001 |
| **Indigenous** | | | | | | | | | | | |
| Beans | 63.30 | 60.43 | 66.07 | 59.01 | 55.44 | 62.49 | 57.52 | 54.05 | 60.92 | 5.110 | <0.001 |
| Vegetables | 30.94 | 28.16 | 33.87 | 33.27 | 30.25 | 36.44 | 36.58 | 33.43 | 39.84 | 4.902 | <0.001 |
| Fruits | 29.64 | 29.94 | 32.50 | 29.91 | 26.60 | 33.44 | 30.66 | 27.34 | 34.19 | 0.246 | 0.844 |
| Fried salty snacks | 14.47 | 12.46 | 16.74 | 17.86 | 15.16 | 20.91 | 14.03 | 12.02 | 16.32 | -0.130 | 0.896 |
| Sweets | 47.83 | 44.66 | 51.01 | 39.47 | 35.77 | 43.30 | 39.37 | 36.05 | 42.79 | -7.630 | <0.001 |
| Soft drinks | 36.98 | 33.97 | 40.09 | 35.04 | 31.66 | 38.57 | 29.24 | 25.83 | 32.90 | -4.787 | <0.001 |

*adjusted for the variables age, gender, macro regions and school administrative status.

for school meals with their own resources or those of their families and purchase food through school cafeterias, which are not necessarily supervised by nutritionists [28, 31].

Fruit consumption fluctuated throughout the survey editions, although in a non-significant way, with a decrease in consumption between 2009 and 2012 and a subsequent increase between 2012 and 2015, reflecting an average variation of 0.149. This difference between the last two surveys (2012 and 2015) can be attributed to the PNAE's implementation in 2009 of the mandatory minimum supply of three servings of fruit or vegetables per week in public

**Table 5. Trends in prevalence of regular food consumption (≥ 5 times/ week) of eating markers, according to school administrative status, from 2009 to 2015.** Brazil, 2009, 2012 and 2015.

| Eating markers | 2009 | | | 2012 | | | 2015 | | | Average variation | p value |
|---|---|---|---|---|---|---|---|---|---|---|---|
| | % | 95% CI | | % | 95% CI | | % | 95% CI | | | |
| **Private** | | | | | | | | | | | |
| **Beans** | 50.14 | 48.80 | 51.48 | 48.62 | 46.72 | 50.52 | 47.61 | 45.47 | 49.77 | -1.719 | 0.002 |
| **Vegetables** | 34.29 | 33.02 | 35.59 | 38.94 | 37.56 | 40.34 | 41.39 | 39.11 | 43.71 | 5.380 | <0.001 |
| **Fruits** | 31.77 | 30.51 | 33.06 | 29.99 | 28.63 | 31.38 | 31.65 | 29.64 | 33.73 | -1.342 | 0.009 |
| **Fried salty snacks** | 14.32 | 13.54 | 15.14 | 16.25 | 15.03 | 17.55 | 13.96 | 12.91 | 15.08 | -1.321 | 0.001 |
| **Sweets** | 49.91 | 48.56 | 51.26 | 39.74 | 38.19 | 41.32 | 41.33 | 39.84 | 42.84 | -9.291 | <0.001 |
| **Soft drinks** | 39.05 | 37.73 | 40.39 | 35.28 | 33.89 | 36.70 | 26.92 | 25.10 | 28.81 | -7.112 | <0.001 |
| **Public** | | | | | | | | | | | |
| **Beans** | 65.83 | 65.11 | 66.54 | 63.89 | 62.79 | 64.98 | 59.48 | 58.40 | 60.56 | -2.716 | <0.001 |
| **Vegetables** | 30.42 | 29.68 | 31.18 | 34.84 | 33.93 | 35.76 | 37.11 | 36.12 | 38.11 | 5.931 | <0.001 |
| **Fruits** | 31.44 | 30.68 | 32.22 | 29.69 | 28.91 | 30.48 | 33.26 | 32.34 | 34.19 | 0.283 | 0.328 |
| **Fried salty snacks** | 11.99 | 11.48 | 12.52 | 15.57 | 14.92 | 16.24 | 14.64 | 13.77 | 15.57 | 1.325 | <0.001 |
| **Sweets** | 51.12 | 50.31 | 51.93 | 43.61 | 42.46 | 44.76 | 41.94 | 40.90 | 42.99 | -6.051 | <0.001 |
| **Soft drinks** | 36.73 | 35.94 | 37.52 | 35.49 | 34.55 | 36.46 | 29.55 | 28.41 | 30.72 | -3.295 | <0.001 |

*adjusted for the variables age, gender, race/skin color and macro regions.

schools' meals [32], whereas consumption varied negatively among private school students. The students enrolled in public schools consume more beans than those enrolled in private schools. In Brazil, there is a public policy that offers food and meals to all students enrolled in public schools, the National School Feeding Program. In 2021, a new resolution will take effect with a proposal to improve the composition of school meals, valuing fresh and minimally processed foods.

The fruit consumption was higher among white adolescents in 2009 and 2012, while in 2015, it was higher among Asian Brazilian adolescents. According to data from the 2004, 2009 and 2013 National Household Sample Survey (Pesquisa Nacional por Amostra de Domicílios —PNAD), black Brazilians had higher eating instability, with limited access to food when compared with white individuals. This can also be influenced by the income, since white Brazilians have a higher monthly income than black individuals [33–35].

Some limitations of this study must be considered. Self-applicable structured questionnaire was used for the interview with the students. This fact debilitates a complex analysis of schoolchildren eating, but such an approach is commonly used in studies for this purpose due to its low cost and amplitude. The time considered in our analyses (6 years), as well as the assessment of food consumption in the last 7 days, may not be enough to detect significant changes in adolescents' consumption. These modifications may be related to changes in price, distribution and access to food. Additionally, the usage of food eating markers is limited because it does not include all foods consumed and that focus only on the frequency of consumption, but it is important to highlight that the evaluation of food consumption in this study used a validated instrument [36]. Another limitation of our study is the existence of missing data, but that does not reach 2.5%.

## Conclusions

To evaluate the three editions of PeNSE enabled us to assess trends of the adolescents' food consumption according to sociodemographic variables, contributing to monitor Brazilian adolescents' health.

The food consumption trends among adolescents was positive, with decreased consumption of unhealthy eating markers and an increase in regular vegetable consumption. Gender, race/skin color and school administrative status influenced food consumption. Despite this positive evolution, the frequency of sweet consumption is still high and higher than that of foods such as fruit and vegetables. It is also worrying that still ¼ of this population consumes soft drinks regularly.

## Acknowledgments

The authors thank the students participating in PeNSE.

## Author Contributions

**Conceptualization:** Hélida Ventura Barbosa Gonçalves, Daniela Silva Canella, Daniel Henrique Bandoni.

**Data curation:** Daniel Henrique Bandoni.

**Formal analysis:** Daniela Silva Canella, Daniel Henrique Bandoni.

**Investigation:** Hélida Ventura Barbosa Gonçalves, Daniela Silva Canella.

**Methodology:** Hélida Ventura Barbosa Gonçalves, Daniel Henrique Bandoni.

**Project administration:** Hélida Ventura Barbosa Gonçalves.

**Software:** Daniela Silva Canella.

**Supervision:** Daniela Silva Canella, Daniel Henrique Bandoni.

**Writing – original draft:** Hélida Ventura Barbosa Gonçalves.

**Writing – review & editing:** Hélida Ventura Barbosa Gonçalves, Daniela Silva Canella, Daniel Henrique Bandoni.

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
