## [Decision Letter · Decision Letter 0]

7 Jul 2020

PONE-D-20-06514

Temporal variation in food consumption of Brazilian adolescents (2009-2015)

PLOS ONE

Dear Dr. Ventura Barbosa Gonçalves,

Thank you for submitting your manuscript to PLOS ONE. After careful consideration, we feel that it has merit but does not fully meet PLOS ONE’s publication criteria as it currently stands. Therefore, we invite you to submit a revised version of the manuscript that addresses the points raised during the review process.

We look forward to receiving your revised manuscript.

Kind regards,

Matias Noll, Ph.D

Academic Editor

PLOS ONE

Additional Editor Comments:

The authors must include in the manuscript and discuss previous recent studies that evaluated this topic:

- https://doi.org/10.1017/S1368980019001010

- https://www.nature.com/articles/s41598-019-43611-x

- doi:10.1017/S1368980018003762

- https://bmjopen.bmj.com/content/6/11/e011571.short

Journal Requirements:

2. Please correct your reference to "p=0.000" to "p<0.001" or as similarly appropriate, as p values cannot equal zero.

3. In your Methods section, please provide additional information about the participants included in your analysis. Please ensure you have provided sufficient details to replicate the analyses such as: a) a description of any inclusion/exclusion criteria that were applied to participant inclusion, b) a statement as to whether your sample can be considered representative of a larger population, c) a participant flowchart.

4. We note you have included a table to which you do not refer in the text of your manuscript. Please ensure that you refer to Table 5 in your text; if accepted, production will need this reference to link the reader to the Table.

Reviewers' comments:

Reviewer's Responses to Questions

**Comments to the Author**

1. Is the manuscript technically sound, and do the data support the conclusions?

Reviewer #1: Yes

Reviewer #2: Yes

2. Has the statistical analysis been performed appropriately and rigorously? 

Reviewer #1: Yes

Reviewer #2: Yes

3. Have the authors made all data underlying the findings in their manuscript fully available?

Reviewer #1: Yes

Reviewer #2: Yes

4. Is the manuscript presented in an intelligible fashion and written in standard English?

Reviewer #1: Yes

Reviewer #2: Yes

5. Review Comments to the Author

Reviewer #1: 5.1. The manuscript is methodologically sound and the data generated supported the conclusions. However, the abstract part lacks brief background information.

5.2. The statistical analysis has been performed appropriately and rigorously.

5.3. All relevant data were included in the manuscript.

5.4. Minor language edition was made to bring the manuscript to the level of high standard.

All other comments were included within the manuscript.

Reviewer #2: This study utilized data from national surveys conducted in three time points to study the temporal variable in specific food consumption. The study utilized a large sample size and analyzed the data appropriately accounting for complex survey design. Below are some comments

1- The authors should explicitly state the study design

2- The authors should comment on the the level of missing data.

3- Line 146: The authors indicated that the outcome variables are percentages. It is not clear what is the numerator and denominator are used in the calculation. A problem of modeling percentages/ proportions is that the model may predict values below 0 or above 100. The authors should justify the use of percentage as outcome variables.

4- Line 158: For the sentence starting "The population aged 14 ...". Are these results for the whole sample or for children aged 14:

5- All tables: Change commas to decimal points in tables. One decimal place will make the numbers in the tables more readable. p-value =0.000 are better written as <0.001

6- Table 2: Clarify the calculations of average variation in the methods

Tables 3-5: Is it trends of food consumption ( (≥ 5 times/ week) ) or trends in percentage of food consumption?

7- Line 227: Does "its" refer to fruits. Please clarify in text as this is a new paragraph

8- Line 127: "during the seven-day period". Please clarify this phrase. Isn't this also a limitation in your study?

6. PLOS authors have the option to publish the peer review history of their article (what does this mean?). If published, this will include your full peer review and any attached files.

Reviewer #1: **Yes: **Dr. Kassa Demissie Abdi

Reviewer #2: No

---

## [Author Response · Author response to Decision Letter 0]

10 Aug 2020

REBUTTAL LETTER

Matias Noll, Ph.D

Academic Editor

PLOS ONE

August 6, 2020

Dear Dr. Noll:

Subject: Submission of revised paper: Temporal variation in food consumption of Brazilian adolescents (2009-2015) [PONE-D-20-06514]. 

Thank you for your email dated on 07 July 2020 enclosing the reviewers´ comments. We thank for their generous comments on the manuscript and have edited the manuscript to submit a revised version of the manuscript that addresses the points raised during the review process. Our responses are given in a point by point bello. Changes to the manuscript are show in a underline/red/bold. 

Editor Comments:

The authors must include in the manuscript and discuss previous recent studies that evaluated this topic: 

- https://doi.org/10.1017/S1368980019001010

- https://www.nature.com/articles/s41598-019-43611-x

- doi:10.1017/S1368980018003762

- https://bmjopen.bmj.com/content/6/11/e011571.short

Response: The indicated articles were included in the article. 

Journal Requirements:

Response: We check all the style requirements of Plos One and, we believe we are suitable. 

2. Please correct your reference to "p=0.000" to "p<0.001" or as similarly appropriate, as p values cannot equal zero.

Response: The suggestion was accepted. 

3. In your Methods section, please provide additional information about the participants included in your analysis. Please ensure you have provided sufficient details to replicate the analyses such as: a) a description of any inclusion/exclusion criteria that were applied to participant inclusion, b) a statement as to whether your sample can be considered representative of a larger population, c) a participant flowchart.

Response: The suggestion was accepted. We reorganized the mentioned text in order to make the inclusion / exclusion criteria of the participants clearer. As well as adding a participant flow figure. 

4. We note you have included a table to which you do not refer in the text of your manuscript. Please ensure that you refer to Table 5 in your text; if accepted, production will need this reference to link the reader to the Table.

Response: We mentioned table 5 in the fifth paragraph of the results. 

Reviewer #1: 

5.1. The manuscript is methodologically sound and the data generated supported the conclusions. However, the abstract part lacks brief background information.

Response: The suggestion was accepted. We correct this point and we add a background information at abstract. 

5.2. The statistical analysis has been performed appropriately and rigorously.

Response: Thank you for your comment.

5.3. All relevant data were included in the manuscript.

Response: Thank you for your comment.

.

5.4. Minor language edition was made to bring the manuscript to the level of high standard.

All other comments were included within the manuscript.

Response: Thank you for your comment.

Reviewer #2: 

This study utilized data from national surveys conducted in three time points to study the temporal variable in specific food consumption. The study utilized a large sample size and analyzed the data appropriately accounting for complex survey design. Below are some comments

Response: Thank you for your comments. 

1- The authors should explicitly state the study design

Response: The suggestion was accepted, and we correct this point. We added figure 1 and, we changed the wording of the study design. 

2- The authors should comment on the level of missing data.

Response: Thanks for the comment. The missing data are a limitation of our study, however these values do not reach 2.5%, as shown in the table below. We inserted this point as a limitation of the study in the discussion. The table below shows the maximum missing for each of the main variables (eating markers foods).

Table 1. Percentages of missing data. 

Variable N. analyzed % of total sample n

Beans 169,737 97.94

Vegetables 169,719 97.93

Fruits 169,702 97.92

Fried salty snacks 169,837 98.00

Sweets 169,687 97.91

Soft drinks 169,709 97.92

3- Line 146: The authors indicated that the outcome variables are percentages. It is not clear what is the numerator and denominator are used in the calculation. A problem of modeling percentages/ proportions is that the model may predict values below 0 or above 100. The authors should justify the use of percentage as outcome variables.

Response: We thank the appointment made by the reviewer. For your analyses we don´t use the proportion as outcome. We build a single database with the three surveys, with year of the survey as a variable. The models were made considering the number of individuals who regularly consumed the different food groups - E.g., the number of adolescents who regularly consume soft drinks as a dependent variable - and the year of the survey (2009, 2012 or 2015) as an explanatory variable, both expressed as continuous variables.

- We agree with the reviewer that it´s not clear in the text and change in the Methods for:

- The statistical significance of the indicator’s temporal trends was assessed using linear regression models with the indicator value as outcome (dependent variable)—e.g., the number of adolescents who regularly consume soft drinks—and the year of the survey (2009, 2012 or 2015) as an explanatory variable, both expressed as continuous variables. The regression coefficient of the model indicates the average rate, expressed in percentage points, of variation of the indicator in the period between the three surveys. All models were adjusted by sociodemographic characteristic (gender, age, macro regions, race/skin color, and school administrative status). The variation corresponding to a regression coefficient statistically different from zero (p value ≤ 0.05) was considered significant.

4- Line 158: For the sentence starting "The population aged 14 ...". Are these results for the whole sample or for children aged 14:

Response: We correct in the text. These results are only for children aged 14. 

5- All tables: Change commas to decimal points in tables. One decimal place will make the numbers in the tables more readable. p-value =0.000 are better written as <0.001

Response: Thanks for the review, we correct this point. 

6- Table 2: Clarify the calculations of average variation in the methods

Tables 3-5: Is it trends of food consumption (≥ 5 times/ week) or trends in percentage of food consumption?

Response: The suggestion was accepted, and we correct this point. The correct is trends in percentage of food consumption. 

7- Line 227: Does "its" refer to fruits. Please clarify in text as this is a new paragraph

Response: The suggestion was accepted, and we correct this point. 

8- Line 127: "during the seven-day period". Please clarify this phrase. Isn't this also a limitation in your study?

Response: Thanks for the comment. PeNSE asks adolescents students about their food consumption in the last seven days prior to answering the questionnaire. Therefore, we do not consider it a study limitation.

We hope that the manuscript is now suitable for publication in Plos One. 

Hélida Ventura Barbosa Gonçalves

---

## [Decision Letter · Decision Letter 1]

2 Sep 2020

Temporal variation in food consumption of Brazilian adolescents (2009-2015)

PONE-D-20-06514R1

Dear Dr. Ventura Barbosa Gonçalves,

We’re pleased to inform you that your manuscript has been judged scientifically suitable for publication and will be formally accepted for publication once it meets all outstanding technical requirements.

Kind regards,

Matias Noll, Ph.D

Academic Editor

PLOS ONE

Additional Editor Comments (optional):

Reviewers' comments:

Reviewer's Responses to Questions

**Comments to the Author**

1. If the authors have adequately addressed your comments raised in a previous round of review and you feel that this manuscript is now acceptable for publication, you may indicate that here to bypass the “Comments to the Author” section, enter your conflict of interest statement in the “Confidential to Editor” section, and submit your "Accept" recommendation.

Reviewer #2: All comments have been addressed

2. Is the manuscript technically sound, and do the data support the conclusions?

Reviewer #2: Yes

3. Has the statistical analysis been performed appropriately and rigorously? 

Reviewer #2: Yes

4. Have the authors made all data underlying the findings in their manuscript fully available?

Reviewer #2: No

5. Is the manuscript presented in an intelligible fashion and written in standard English?

Reviewer #2: Yes

6. Review Comments to the Author

Reviewer #2: All comments were addressed.

The authors addressed my concerns related to statistical analysis and study design.

7. PLOS authors have the option to publish the peer review history of their article (what does this mean?). If published, this will include your full peer review and any attached files.

Reviewer #2: No

---

## [Editor Report · Acceptance letter]

4 Sep 2020

PONE-D-20-06514R1 

Temporal variation in food consumption of Brazilian adolescents (2009-2015) 

Dear Dr. Ventura Barbosa Gonçalves:

I'm pleased to inform you that your manuscript has been deemed suitable for publication in PLOS ONE. Congratulations! Your manuscript is now with our production department. 

Kind regards, 

on behalf of

Dr. Matias Noll 

Academic Editor

PLOS ONE